# Silica Gel Chromatographic Methods for Identification, Isolation and Purification of Gossypol Acetic Acid

**DOI:** 10.3390/gels10070432

**Published:** 2024-06-29

**Authors:** Amro Abd Al Fattah Amara, Mohamed Hesham El-Masry, Gamal Ali Salem, Hoda Hassan Baghdadi

**Affiliations:** 1Protein Research Department, Genetic Engineering and Biotechnology Research Institute, City of Scientific Research and Technological Applications (SRTA-City), Alexandria P.O. Box 21934, Egypt; 2Biotechnology Department, Graduate Studies and Research Institute, Alexandria University, Alexandria P.O. Box 21526, Egypt; 3Environmental Science Department, Graduate Studies and Research Institute, Alexandria University, Alexandria P.O. Box 21526, Egypt; gsalem662000@yahoo.com (G.A.S.); hodabaghdadi@yahoo.com (H.H.B.)

**Keywords:** cottonseed, silica gel, silica gel column, robust spherical silica gel, gossypol, gossypol acetic acid

## Abstract

Several cottonseed varieties are cultivated in different countries. Each variety produces a different amount of gossypol as a natural toxic compound. The rising interest in cottonseed products (oil and feed) increases the demand for establishing simple methods for gossypol detection. Silica gel-based methods are ideal for its isolation, purification, and characterization. Silica gel-based methods are variants and can be used as simple methods for tracking plants’ compounds. In this study, gossypol was isolated, characterized, and purified as gossypol acetic acid in the form of yellow crystals. Methods used for its characterization were TLC, preparative TLC, silica gel column, UV/IR spectrophotometer, and HPLC (robust spherical silica gel). A comparative study between its amount in both the Egyptian and Chinese varieties was performed. Under the experimental conditions, the Egyptian’s cottonseed contains 8.705 gm/kg, while the Chinese’s cottonseed contains 5.395 gm/kg. The TLC used in this study proved to be fast, accurate, and inexpensive. It can be used for gossypol acetic acid evaluation and quantification. Additionally, using TLC as a pre-purification step will give a pre-judgment for the sample’s purity and quality. This step will protect the expensive HPLC silica gel-based column from any unexpected impurities. During each step, the silica gel itself could be simply removed by paper filtration. Collectively, the different silica gel-based methods as well as the other used methods are recommended for better Gossypol acetic acid isolation, purification, and characterization, as well as for maintaining HPLC columns.

## 1. Introduction

Cottonseed has economic importance as a source of oil and as animal feed. There are a number of existing cottonseed varieties. Even though the seeds have high oil and nutritive components, unfortunately, a natural toxic pigment exists. It is given the name “gossypol” (related to the plant name, “*Gossypium barbadense*” in the family Malvaceae) [1]. Thus, removing gossypol or degrading it is an important step [2,3,4,5,6]. So, there is a real need for a fast, cheap, and practical method(s) for its detection, isolation, purification, crystallization, identification, and quantification.

Gossypol is a non-lactonic sesquiterpene. It has different physiological and biological activities. It is usually prepared as gossypol acetic acid (GAA). GAA shows a growing interest because of its biological properties, such as anticancer [7], antiviral [8,9], antimicrobial, and antifertility properties [10,11,12]. It can inhibit cellular macromolecules [13]. The crude extract of the cottonseed (containing gossypol) could inhibit the formation of crown gall tumors in the potato disc bioassay, which was induced by *Agrobacterium tumefaciens* [14], and could inhibit hepatic microsomal benzo(α) pyrene hydroxylation and hydrogen peroxide production in mice treated with lindane [15].

Gossypol has shown a mean treated over-control tumor volume ratio (T/C) of 150% at 10 mg/kg in the PS system [16]. It has a unique chemical structure comprises the following formula: 1,1′,6,6′,7,7′-Hexa hydroxy-3,3′-dimethyl-5,5′-bis (1-methylethyl) [2,2′-binaphthalene] -8, 8′-dicarboxaldehyde; 1,1′,6,6′,7,7′-hexahydroxy-5,5′-diisopropyl-3,3′-dimethyl [2,2′-binaphthalene]-8, 8′-dicarboxaldehyde; 2,2′-bis [1,6,7-trihydroxy-3-methyl-5-isopropyl-8-aldehyde naphthalene]; 2,2′-bis [8-formyl-1,6,7-trihydroxy-5-isopropyl-3-methyl naphthalene]. It has a general structure of C_30_H_30_O_8_, with a molecular weight of 518.54 [17]. In the case of GAA, its general structure is C_30_H_30_O_8_.C_2_H_4_O_2_, with a molecular weight of 578.62. Several optimizations for gossypol extraction were performed over the years; for example, Pelitire et al. (2014) revised different solvents for gossypol isolation [18].

Its toxicity was investigated on different cancer cell lines, such as Her2-positive breast cancer cells [19]. It is able to cause up-regulation of somatostatin receptors 2 and 5 in DU-145 prostate cancer cells [20] and cultured murine erythroleukemia cells [21]. It can inhibit the growth of colorectal cancer, hepatocellular carcinoma [22], and the like. It has also been demonstrated that gossypol inhibits proliferating Ehrlich ascites tumor cells [16]. (±)-Gossypol induced apoptosis and autophagy in head and neck carcinoma cell lines [23]. Gossypol decreased cell viability and downregulate the expression of some genes in human colon cancer cells [24]. Gossypol was demonstrated to be a specific inhibitor of DNA synthesis [25]. Rosenberg et al. (1986) highlighted that gossypol affects nuclear DNA by upregulating DNA replication and mismatching proteins [26]. The mechanism of the anticancer activity of gossypol is to induce apoptosis through the suppression of anti-apoptotic proteins of the Bcl-2 family [27].

Gossypol has antiproliferative and antimetastatic effects on Mat-lylu prostate cancer cells. It can be a potential therapeutic agent for androgen-independent human prostate carcinoma [28]. Oral gossypol is reported to be safely used on an outpatient basis for the treatment of metastatic adrenal cancer [29]. The genotoxic effect of GAA was evaluated by determining the frequency of micronuclei and mitotic index in male mouse bone marrow cells in vivo [30]. The rate of DNA degradation by the gossypol–Cu^+2^ complex was found to be the same both in the presence and absence of molecular oxygen [31]. It is reported as an inhibitor for Bcl-2, Bcl-XL, Bcl-W, and Mcl-1 [32]. For more details refer to Fulda (2010) and the references within [32]. For that, it and its derivatives are promising candidates for cancer treatment [7,32,33,34].

There is an increasing interest in using gossypol as an antivirus [35,36,37,38,39,40]. Gossypol has many other important effects, where both racemic mixtures and enantiomers of gossypol inhibit replicating human immunodeficiency virus-type 1 (HIV-1) [41,42]. The in vitro anti-amoebic effects of gossypol were reported against axenic trophozoites from five *Entamoeba histolytica* strains [43].

The science of chromatography is the collective term for a set of laboratory techniques for the separation of mixtures into their components. They all have a stationary phase (a solid or a liquid supported by a solid) and a mobile phase (a liquid or a gas).

The great movement from the close column to the open column (thin layer chromatography) significantly solved many problems concerning the mobile phase movement. In 1938, Izmailov and Shraiber [44] described the basic principle of the procedure in an article entitled “Analysis by Drop chromatography and its Application in Pharmacy” [44]. They applied the method to the separation and characterization of extracts of medicinal plants (tinctures of the Soviet Pharmacopoeia VII). Zechmeister has early highlighted that in 1938 “The chief problem is not the development of suitable apparatus (referring to column chromatography) but the precise identification of the adsorbed substances” [45]. The principle of TLC is the distribution of a compound between a solid fixed phase applied to a glass or plastic plate (aluminum oxide or silica gel) and a liquid mobile phase. The solubility rule “Like Dissolves Like” is followed, where the mobile phase will carry the most soluble compounds the furthest up the TLC plate. The compounds that are less soluble in the mobile phase and have a higher affinity for the particles on the TLC plate will stay behind. These properties give the TLC great importance in isolating enantiomers and racemic mixtures [46]. The broad range of stationary phases now available in TLC has made this technique more generally applicable and competitive with HPLC. For more details about TLC, refer to Stahl 1969 and the references within [47].

Silica gel can be used in different forms and combinations with other molecules for isolating botanical compounds. One of its unique properties is its resistance to chemical changes in the presence of different solvents. In contrast to other forms of gels, such as the gel groups that are used to isolate proteins, silica gel is more stable and can be easily retrieved from a column or from the glass plate surface; it can also be cleaned, dehydrated, and reused. Its chemical properties enable its coexistence with simple assistant compounds like starch, which was used for preparing TLC. TLC is extremely useful in the analysis of natural products; it plays an important role in the isolation and purification of different metabolites; and it is a rapid and simple tool.

One interesting property of the GAA is its different chemical isomers and enantiomers, like (+)-gossypol triacetic acid or (−)-gossypol triacetic acid [48,49,50]. The presence of enantiomers is a major concern in gossypol isolation as GAA. The enantiomers have different biological activities and could give incorrect results if they were not isolated and separated from each other [43,51,52].

Some compounds, reported by different research groups or from different plants, obviously represent the same structures. Meanwhile, they showed different R_f_ values on thin layer chromatography (TLC) [53,54].

This study is concerned with simplifying different silica gel chromatographic based methods, including TLC, preparative TLC, column chromatography, and HPLC (robust spherical silica gel). Other simple analytical methods, such as UV/infrared spectrophotometric analysis, are included. An authentic GAA sample was used in the entire study to assist in quantifying and characterizing the prepared GAA. The silica gel-based methods and the other methods used are recommended for fast isolation, characterization, and purification. Different silica methods give different choices. In the present study, the methods used ranged from simple, handy, in-house, and inexpensive to highly accurate and sophisticated.

## 2. Results and Discussion

Cottonseed from Egypt was collected, ground, and sieved into fine particles to enable better extraction. The grinding step is essential for removing debris and facilitates the extraction process. For defatting the seed materials, petroleum ether was used. After several extraction steps with the ether, the seeds were air-dried to remove any residual solvent. After that, diethyl ether was used to extract the gossypol. The diethyl ether extract was reduced to concentrate the sample at 40 °C, which was used during the entire step to avoid degrading the gossypol or GAA. For that, the solvent evaporation was performed under vacuum.

For its simplicity and efficacy, we aimed to select some of the simplest and most direct silica gel-based methods, either from the research previously conducted on gossypol or from other phytochemistry protocols. The study introduces a straightforward protocol for isolation, purification, and characterization of GAA. Generally, the methods described in the current study were collected based on their simplicity compared to some other tested methods and based on our scientific group experiences in the field of phytochemistry.

Simple silica gel methods could assist in better purification. That will protect expensive columns from any unexpected impurities. In addition, using the described method as a model could help in isolating pure compound(s) from plant source(s). This will assist in isolating another interesting natural product, particularly those that could be isolated using solvents and purified using different silica gel methods.

Simple methods for purification and recrystallization were used. Multiple runs represent washing with petroleum ether, followed by filtration, and then re-dissolving in diethyl. Finally, acetic acid was used to precipitate the gossypol as GAA. For further purification, TLC preparative chromatography was used. TLC (silica gel G60) preparative chromatography ensures an additional purification step to obtain a suitable quantity in a short time. Samples were taken and tested for their presence or to evaluate their purity in any of the above-described steps, as shown in Figure 1a,b. From Figure 1, one could observe a well-separating process where all the separated points are in the same line (A, B, and C). That indicates a well-saturated chamber, as described by Stahl (1969) [47]. The obtained R_f_ for spot C = 1.5/11 = 0.136, R_f_ for spot B = 3.5/11 = 0.318, and R_f_ for spot A = 7/11 = 0.636. The importance of the TLC in the separation of enantiomers was revised by many authors. For more details, refer to Mack and Hauck (1988) and the references within [52],

Acetic acid was used to precipitate the gossypol as GAA. To our knowledge, the precipitation occurs after putting the sample in a tightly closed bottle under static conditions in a dark chamber. Apparently, darkness was an essential factor in precipitating gossypol as GAA. The GAA was separated on the TLC into two bands, A and B. Another band, C, appeared at the bottom of the TLC beside the starting point (which was neglected), as shown in Figure 1b. Apparently, that could be explained because of the presence of racemic GAA, (+)-gossypol triacetic acid, or (−)-gossypol triacetic acid. As well, gossypol has more isomers and enantiomers. For more details, refer to Dowd et al. (1999) [49], Dowd (2003) [48], and Yildirim-Aksoy (2004) [50]. The concentrations of GAA present in spots A and B after preparative TLC chromatography using HPLC analysis (Figure 2) indicate that spot A contained 17.5% of GAA while spot B contained 1.02% GAA. Meanwhile, the GAA in spots a and b did not have identical configurations, as explained by Dowd et al. (1999), Dowd (2003), and Yildirim-Aksoy (2004) [48,49,50]. That could explain its presence in different spots because of different mobility on the TLC. Bereznitski et al. (2001) reported that conducting chiral separation using TLC has generally some major advantages, such as being more flexible with high throughput than HPLC and able to use different types of interactions that produce critical separation. In this study, visualization, quantification (scratching followed by isolation, purification, and weighting), and isolation of a considerable purified amount using preparative TLC are additional privileges to the process. In fact, other issues could be included, like time, cost, material recovery, etc. [51]. Visualizing different spots isolated for a compound and giving the same optical spectra will be a strong indication of the presence of isomers. Bereznitski et al. (2001) reported that: “The process of enantiomeric separation by thin-layer chromatography (TLC) alleviated” [51]; “the separated enantiomers could be visualized directly on the plate” [51].

Mack and Hank (1988) reported that the mechanisms of chiral recognition can be subdivided into five groups: (1) formation of inclusion complexes; (2) charge-transfer complexation with Pirkleo-type chiral stationary phases (CSP); (3) formation of diastereomers; (4) selective interactions with organic polymers; and (5) chiral ligand exchange [52].

Selective interactions with organic polymers, as reported by Bereznitski et al. (2001), were previously described. Dalglish (1952) postulated his “three point rule” for chiral recognition in the chromatographic separation of optical antipodes [55]. Stahl (1969) reported that starch, a polysaccharide, is suitable as an adsorbent for TLC. It contains many hydroxyl groups and is consequently strongly hydrophilic [47].

TLC can usually be used as a pre-investigation step before the HPLC, as highlighted by Aboul-Enein et al. (1999): “TLC, however, can be used to screen large numbers of compounds and solvent systems for HPLC and often produces unusual enantioselectivities” [56]. Gossypol exists mostly in two enantiomeric (isomeric) forms designated plus (+) and minus (–), and *Gossypium* spp. produce both forms in varying proportions [57]. Isolating racemic mixtures and enantiomers of gossypol is of great importance, as they were reported to inhibit replicating human immunodeficiency virus-type 1 (HIV-1) [41,42].

The precipitated gossypol as GAA was subjected to a series of washing steps. The GAA that was obtained from the precipitation and recrystallization process was about 2.6 g/kg cottonseed, and it was in the form of brilliant yellow microcrystals (0.26% based on powdered cottonseed). The difference in the calculated amount between the TLC preparation steps and that obtained from the HPLC data (as below) proves the efficacy of the HPLC in the accurate determination of the gossypol quantity. Meanwhile, during the crystallization, purification/recrystallization steps, a loss in quantity is expected. One should differentiate between the quantitative determination of the existing amount, which needs a sophisticated instrument like the HPLC, which needs an amount in µg (or less), and the need for purifying a suitable amount of gossypol (as GAA) to conduct certain experiment(s).

Silica gel column chromatography is another alternative purification method used. Even the gossypol is dissolved in 70% aqueous acetone, but upon loading the sample into the silica gel column, because of its absorbency, it enables the sample to be trapped and adsorbed in the column. The silica gel column will perform more than one function to purify the gossypol. First, it will prevent passing any solid material. Second, it will enable the use of a concentrated sample. Third, it will enable fast washing using a minimum amount of 100% petroleum ether. Fourth, the sample could be simply eluted using the same solvent used to dissolve the gossypol (70% aqueous acetone).

A simple spectrophotometric method was adjusted to quantify GAA using a spectrophotometer. An authentic sample from Sigma was used as a reference. GAA spectra were detected using wavelengths in the range of 188 and 500 nm. The best wavelength in the peaks obtained from UV spectra was at 440 nm, as shown in Figure 3. A curve was generated from the authentic sample, and the purified gossypol was evaluated. Furthermore, the spectra of the authentic sample were compared with those of the purified GAA and proved to be identical, as shown in Figure 2a,b. The results obtained either from the standard curve or from the spectra prove that we have a well-purified compound. IR analysis was further used to compare the purified GAA with the authentic sample, as shown in Figure 4. The spectra of both are identical.

The infrared spectrum of gossypol using a potassium bromide disk showed the following absorption bands (Figure 4): the broad bands of strong intensity at 3500 and 3420 cm^−1^ are attributed to OH stretching vibration. Bands of moderate intensity at 2960, 2920, and 2860 cm^−1^ are ascribed to CH stretching vibration. The band of strong intensity at 1700 cm^−1^ is because of CHO stretching vibration. Two bands at 1440 and 1380 cm^−1^ are because of C–H banding. The band at 1380 cm^−1^ is because of CH_3_ symmetrical deformation. The band at 1340 cm^−1^ is because of the OH band. Bands characteristic of C–OH appeared at 1240 and 1180 cm^−1^. Bands below 1120 (1100, 1050, 960, 840, and 700 cm^−1^ represent the “finger print’’ of GAA.

The melting points of the crystalline substance and the reference sample of GAA were the same. The melting points were 184 °C. Even though it is a simple experiment, it could prove or disprove the identity of two compared samples. The concentration of gossypol varies from one variety to another. For comparison of the gossypol obtained from the Egyptian cottonseed and the Chinese cottonseed, HPLC was used. The results indicated that the yield of gossypol from the Egyptian cottonseed was 8.705 gm/kg, while the yield produced from the Chinese cottonseed was 5.293 gm/kg.

The methods used for identifying gossypol and GAA were simplified in this study (Figure 5). We recommend the use of the different silica gel methods and the analyses described in this study for evaluating the gossypol and GAA in cottonseed oil and animal food products. It might be useful to use cottonseed varieties with less gossypol content on one side [18]. On the other side, cottonseed with a high gossypol content is more resistant to different infections [14]. Performing chiral separation using TLC had three major advantages. First, the detection of the analytes in TLC is more flexible than in HPLC. Second, the sample throughput in TLC is higher than in HPLC, and third, the possibility of using different types of interactions in two-dimensional TLC produced marvelous separations.

## 3. Conclusions

The paper summarizes some experiences in the fields of phytochemistry and chromatography. It combines the real use of silica gel to produce GAA as a purified product with describing different experiments concerning its quantification, identification, and validation against an authentic sample. From the first point, an authentic gossypol acetic acid sample was used to assist the whole study. The paper contains many details that could help in identifying other compounds from plant extracts. TLC, as proved in this study, has such unique properties that it could isolate different enantiomers, which agrees with many other previous studies. It is a simple protocol that can be controlled by the naked eye, and any separated compound can be re-separated and repurified. That will significantly assist any biological or chemical studies concerning plant compounds. Beside the TLC, different kinds of silica gel-based methods are used, such as silica gel column chromatography and the HPLC. Non-intentionally, each of the tested compounds and the used tools and experiments validate each other (Figure 5), and the authentic sample controls the overall success of the process. On the one hand, the GAA, which contains enantiomers or, in addition, might contain a non-complete structure from the gossypol synthesis pathway, was separated into three bands on the TLC. On the other hand, the authentic sample matches perfectly both the spectrophotometer results and the IR result.

The main aim of this study is to enable scientists to purify gossypol as GAA using simple and efficient silica gel-based methods to use it in different experiments. Meanwhile, it is showing the power of silica gel-based techniques. What is the most important issue in the present investigation? It is the up-front method for isolating gossypol until its characterization and quantification, which could stand alone as a protocol for preparing gossypol as a GAA in the lab. This will enhance its investigation of many cancers’ cell lines and other biological interests. In this study, even TLC proved to be efficient in purifying the GAA and giving a visual judgment, while the HPLC should be used for accurate quantification of the amount of gossypol that existed in the cottonseed in different variants (e.g., Egyptian and Chinese). For the general preparative method for GAA, the use of TLC, or column chromatography, is recommended. The other spectrophotometric methods (UV/IR) could be helpful in other experiments for fast identification and quantification. TLC in this study proves to be able to separate GAA (proposed to contain enantiomers) in the form of different bands, as shown in Figure 1, which cannot be detected spectrophotometrically or by IR. In addition, each band can then be repurified and recrystallized to obtain a pure form of GAA. This study gives a simple solution for detecting different configurations for a single compound isolated from the same source with similar isolation and purification conditions and shows different R_f_ on TLC, as shown in Figure 1, and the possibility of purifying each one (from each spot) alone.

## 4. Materials and Methods

### 4.1. Cottonseed Sample

Gossypol was extracted and purified as GAA from Egyptian cottonseed. The used seeds were obtained from Alexandria Oil and Soap Company (Alexandria, Egypt).

### 4.2. Preparation of Seed Material

Three kilograms of dried cottonseed was ground in a blender using standard criteria. The grinded material was passed through a coarse sieve to remove larger parts of the lint and hulls. Two kilograms of the powdered seeds was transferred into a suitable, clean, and dry glass container.

### 4.3. Extraction with Petroleum Ether/Diethyl Ether

Two kilograms of the powdered seeds was first extracted with 3 L of petroleum ether in a shaker (100 rpm) at room temperature using standard criteria. The resulting extract was filtered through filter paper (Whatman N°1). The extraction process was repeated five times to remove the oily component of the powdered seeds. A watch glass was used to control the success of the process, where about 250 µL of the solvent was loaded on its surface and left to be evaporated. The successful process shows no residue on the watch glass surface (transparent). The powdered seeds were air-dried at room temperature to get rid of any solvent. The oil-free seed material was then extracted with 3 L of diethyl ether. The extraction step was further carried out until no change in the color of diethyl ether was observed. Watch glass was used to control the process as described above. The filtrates were concentrated under vacuum using a rotatory evaporator at 40 °C.

### 4.4. Precipitation of Gossypol as GAA

The concentrated extract obtained from a 2 kg sample was transferred into a conical flask, and then the mixture was treated with 40 mL of glacial acetic acid added dropwise until pH 3 was obtained. The flask was well closed and placed in a dark, static place (for one week), where yellow crystalline precipitate was obtained at the bottom of the flask. The supernatant liquid was discarded, and the precipitate was taken and dried in a desiccator [1].

### 4.5. Purification and Recrystallization

The precipitate was washed 3 times with 1 mL of glacial acetic acid, followed each time with 10 mL of petroleum ether. Each time, the mixture was filtered through filter paper (Whatman N°1) and finally, the precipitate was dissolved in 5 mL of diethyl ether containing 1 mL of glacial acetic acid. The ether was followed by evaporation until the purified crystals began to be readily separated. The crystals were dried in a desiccator at room temperature and then weighted.

### 4.6. Detection and Monitoring GAA by TLC

One milligram of each of the crystalline precipitate and reference GAA (Sigma) was separately dissolved in 50 μL of chloroform in a clean and dry glass container. Each sample was separately spotted at the starting point (baseline) across the chromatoplate [(20 × 20 cm) coated with 0.5 mm thick 0.25 silica gel G60 containing 0.6% starch (Merck)]. The chromoplate was previously activated by heat using standard criteria. The chamber was installed in a static place at room temperature (about 20 °C), and the running solvent system was loaded inside. The chamber was closed for 15 min or until saturation. The activated chromoplate was cooled in a 37 °C incubator and left at room temperature for about 5 min. The samples were loaded on the starting line (starting point), as shown in Figure 1. After 3 min, the plate was loaded into the saturated chamber, where the solvent system was brought down to the starting line. The samples were vertically migrated using the solvent system: ether:petroleum ether (1:1) as a running solution. After enough migration and separation, the plate was allowed to dry, and the separated spots could be seen under the visible light as yellow spots (Figure 1) [58]. The TLC was used to monitor GAA in the different samples during the preparation and purification steps (Figure 1). The R_f_ value for each spot was calculated using the formula: R_f_ = distance traveled by component/distance traveled by solvent

### 4.7. Preparative TLC for Purification of GAA

Thirty milligrams of the crystalline precipitate was dissolved in 300 μL of chloroform in two clean and dry (as a duplicate replica) glass containers using standard criteria. The solution was separately streaked in the form of a band across the baseline of the silica gel G60 chromatoplate (containing 0.6% starch as above), and then the chromatoplate was processed using the solvent system (ascending development) ether: petroleum ether (1:1). Alternatively, dense spots were loaded in the form of separated spots on the base line, as shown in Figure 1a.

The plate was air dried, where two separate bands (a and b) appeared under visible light. Another band (C) on the baseline was neglected (as in Figure 1). Images were taken for visualization. Each band was separately collected and then eluted with ether, until no residue was obtained upon evaporating a small volume of the colorless eluate to dryness on a watch glass (clean and dry). Each eluted band was filtered, and the filtrate was concentrated under vacuum to about 2 mL, then left to crystallize. Finally, each crystallized material (a and b as in 4.5) was dried in a desiccator and then tested for the purity of the GAA using each of the spectrophotometer and HPLC (as below).

### 4.8. UV Spectrophotometer Analysis for the Spectra of GAA

The crystalline precipitate from petroleum ether–diethyl ether was dissolved in 95% ethyl alcohol and then determined spectrophotometrically. Forty micrograms of both the GAA precipitate and reference sample were diluted each with 3 mL of ethyl alcohol and scanned at 500–188 nm using a quartz cuvette to determine the presence of GAA in the precipitate [59]. The obtained spectra of both the GAA precipitate and reference sample are visualized in Figure 2. The absorbance of the top of the best clear band (at 440 nm) was used to measure GAA spectrophotometrically in further investigations.

### 4.9. GAA Standard Curve

Two milligrams of standard GAA (Sigma-Aldrich-USA, St. Louis, MI, USA) were weighed accurately in a clean and dry glass tube and were then dissolved in 1 mL of absolute ethanol. Volumes of 20, 40, 60, 80, 100, 120, 140, 160, and 180 μL of GAA were used as a stock solution. Zero absorbance was adjusted against an absolute ethanol blank at 440 nm. The absorbency of the prepared dilutions was determined with a PYEUnicuan spectrophotometer. The standard curve was generated using standard criteria.

### 4.10. Determination of the GAA (Separated by TLC) Using HPLC

Bands A and B from the TLC chromatoplate were removed and then separately treated by chloroform to dissolve the separated substance. The chloroform mixture was filtered to remove the silica–starch layer. The filtrate was further evaporated to obtain the purified substance in a crystalline form. Briefly, 1.9 mg of the purified crystals obtained from band A and 1.5 mg of band B were separately dissolved in 10 mL of 70% aqueous acetone. Five microliters of each was separately injected in HPLC column (Column Art. 720023 ET 250/814 NUCLEOSIL^®^ 10 C18 MACHEREY. NAGEL, Yokosuka, Japan) to detect GAA. The column contains robust spherical silica gel, hydrophobic, weakly polar, and end-capped phases for RP chromatography. The separation mechanism was based on hydrophobic (van der Waals) and slight residual silanol interactions. The following were the conditions under which the apparatus was adjusted: a UV detector (365 nm), at 35 °C, a mobile phase of 0.15 M phosphoric acid in acetonitrile/water (4:1), a flow rate of 1.5 mL/min, and a speed of 5 mm/mL

### 4.11. Silica Gel Column Isolation and HPLC Quantification of Gossypol from Egyptian and Chinese Cottonseeds

One gram of finely powdered Egyptian and Chinese cottonseeds was separately extracted from a 10 mL solution of 70% aqueous acetone in a clean and dry glass container and then incubated with shaking (150 rpm) at 25 °C for 6 h. The extract was clarified by centrifugation at 4000 rpm, and the supernatant was kept in a clean and dry glass container. Silica gel column chromatography was used for further purification (Merck G120 column 15 cm × 1 cm). The 70% aqueous acetone supernatant was allowed to be adsorbed on the silica gel column and then washed with 100% petroleum ether to remove any impurities. The purified gossypol in the column was then eluted with 100 mL of 70% aqueous acetone.

### 4.12. HPLC Method for Analysis of GAA in Egyptian and Chinese Cottonseeds

Briefly, 1.5 μL from the eluent was taken for detecting gossypol using HPLC (under the same conditions as above). Qualitative and quantitative results of GAA in different samples were obtained by comparing the peak retention time and peak area of the standard solution and sample extract under the same conditions of operation (as above) [2,60,61].

### 4.13. Melting Point of GAA

An electrically heated melting point apparatus was used to study the melting point of any crystalline substance. A few crystals of the crystalline precipitate were introduced into the melting point tube and then exposed to electrical heat. The reading of the thermometer was taken at the point at which the crystalline precipitate melted. The same procedure was used with a reference sample of GAA (Sigma Co., Sydney, Australia).

### 4.14. Infrared Spectral Measurements for GAA

Infrared spectra in a range of 600–4000 cm^−1^ were measured with a Perkin-Elmer Model 1420 spectrometer (Connecticut, CT, USA). The KBr pellets needed for IR analysis were prepared with the precautions recommended by O’Conner et al. (1954) [62] to remove moisture. Each sample (0.8 mg) was mixed with 300 mg of KBr. The mixture was placed in an agate mortar, and after being grinned manually for several minutes, the mixture was transferred to the KBr disk press, which was evacuated for 15 min (pressure of 11 tons/cm^2^) [62,63].

## Figures and Tables

**Figure 1 gels-10-00432-f001:**
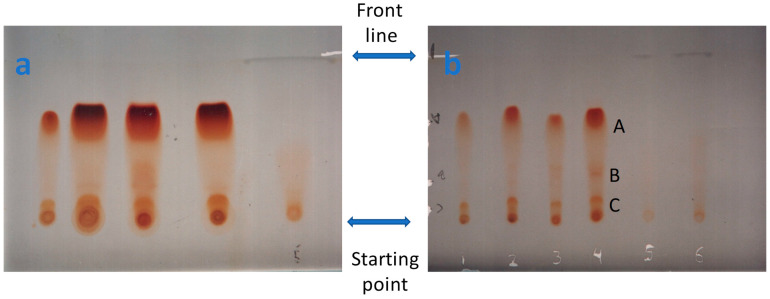
(**a**); Different TLC samples (A, B, C) showing different amounts of gossypol; acetic acid (samples collected during the purification steps), (**b**); purified samples show 2 bands A and B.

**Figure 2 gels-10-00432-f002:**
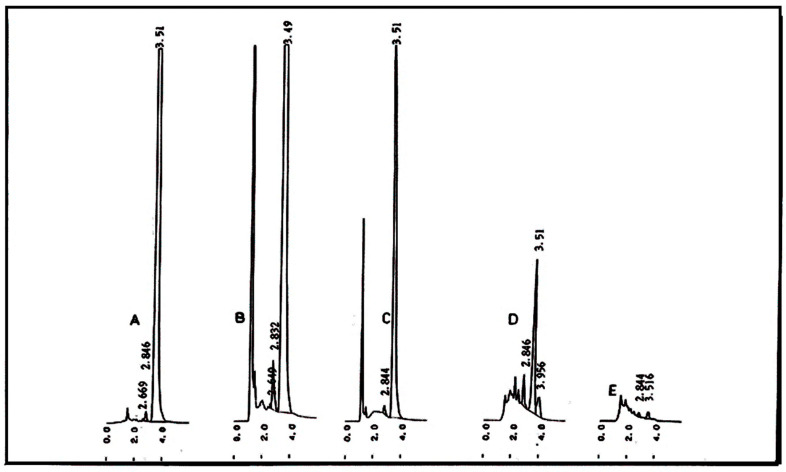
HPLC pattern of A—authentic sample; B—Egyptian GAA sample; C—Chinese cottonseed gossypol; D—crystalline purified precipitate from TLC spot (Figure 1a), and E—crystalline precipitate gossypol from TLC spot (Figure 1b).

**Figure 3 gels-10-00432-f003:**
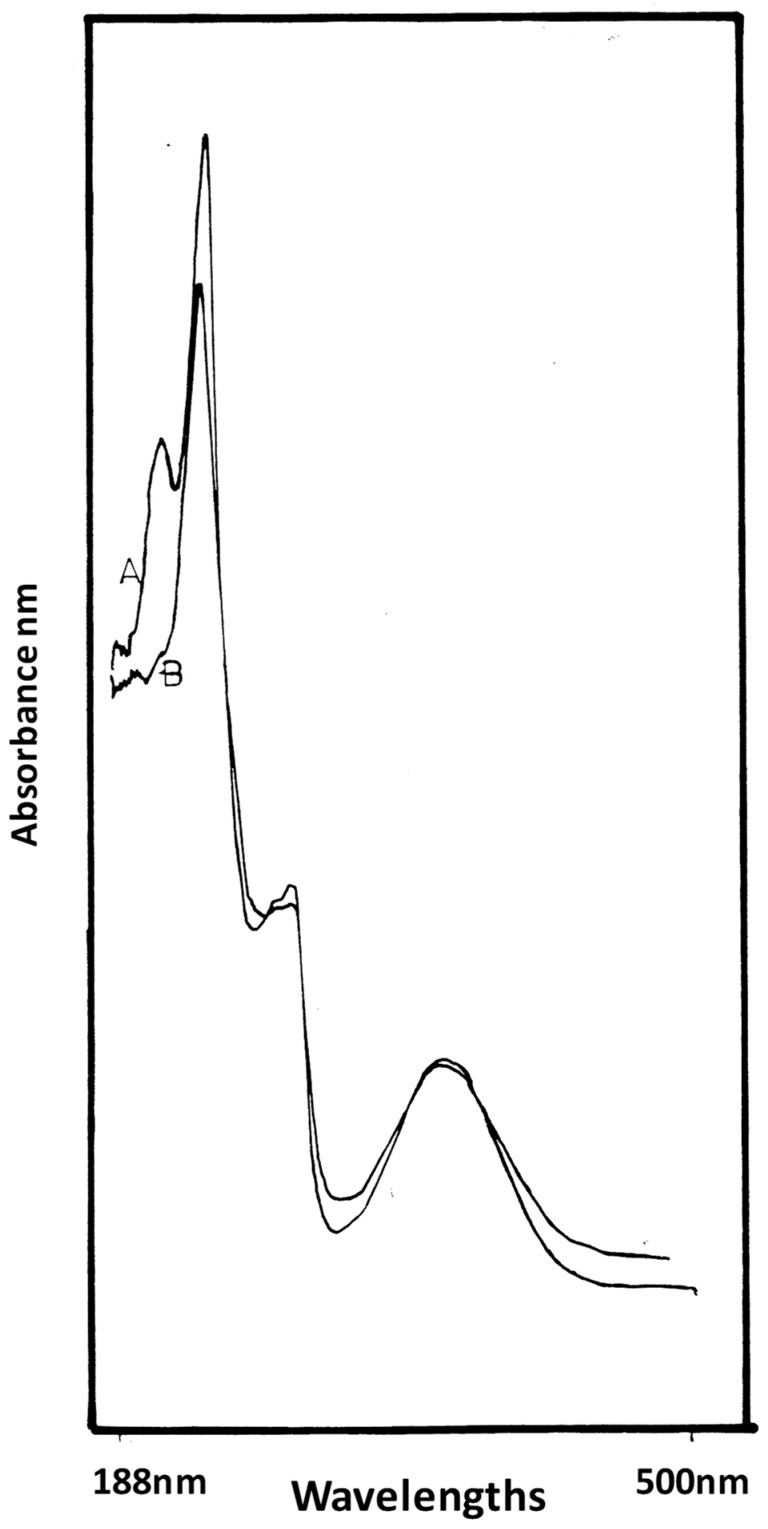
UV spectrum of GAA reference sample (A); Isolated crystals (B).

**Figure 4 gels-10-00432-f004:**
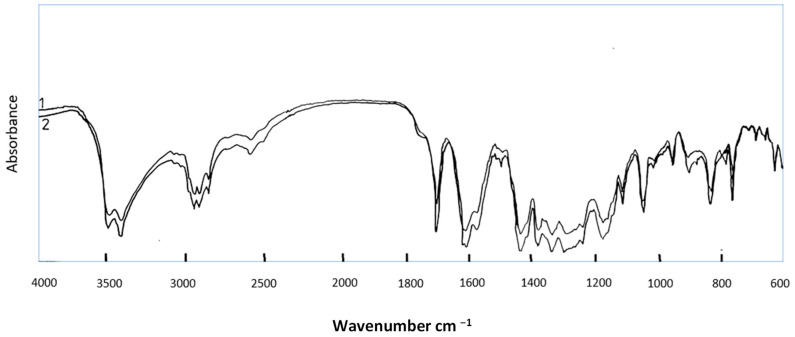
IR spectrum for (1) GAA reference sample; (2) GAA purified samples.

**Figure 5 gels-10-00432-f005:**
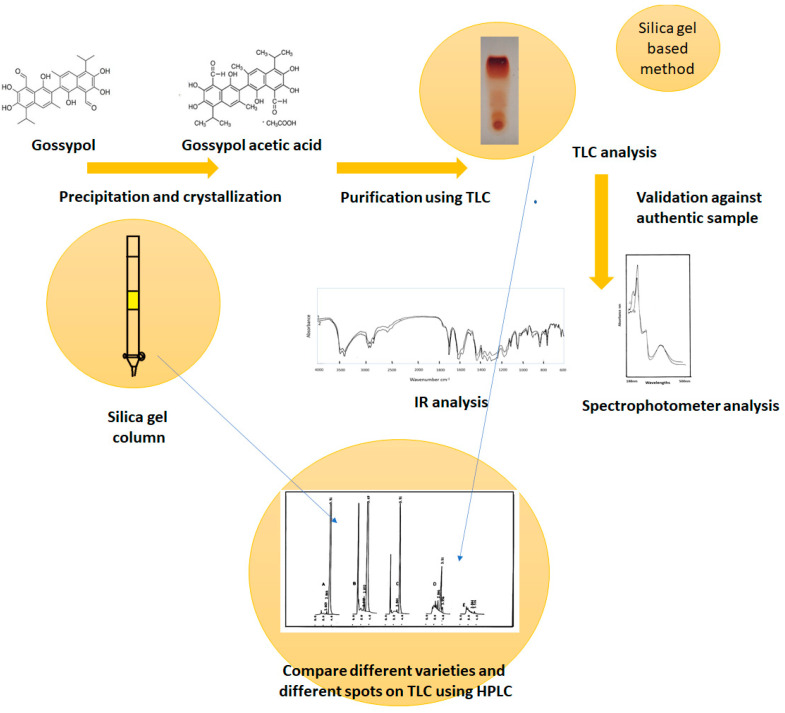
The diagram shows the overall process where gossypol is isolated as GAA crystals, TLC was used to analyze its purity, IR and spectrophotometric analysis its identity against authentic sample proves successful crystallization to GAA, silica gel column chromatography used for preparing the sample for the HPLC. HPLC is used to compare between the Egyptian and Chinese cottonseed varieties. TLC, silica gel column chromatography, and HPLC are different tools for analysis that use silica gel as a stationary phase. Only silica gel-based method could detect the presence of enantiomers visually.

## Data Availability

The data presented in this study are openly available in article.

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
