# Peer review of "Silica Gel Chromatographic Methods for Identification, Isolation and Purification of Gossypol Acetic Acid"

_gels, 2024, doi:10.3390/gels10070432_

Round 1
Reviewer 1 Report
Comments and Suggestions for Authors
The authors attempted to identify, isolate and purify one of the natural toxic compounds, i.e. gossypol, present in cottonseed. They used some chromatographic methods and also UV/IR analysis. Additionally, they studied the similarity between Egyptian and Chinese varieties of cottonseed. The derivative, i.e. gossypol acetic acid, has many pharmacological applications, so it is significant in pharmacy. It also comes from cottonseed, which the authors write about extensively in the introduction to the publication.
The methodology thoroughly describes sample preparation and each point of analysis in detail.
However, I have some comments:
- on what basis did the authors determine that the spots (Fig. 1) contain 17.5% and 1.02% of GAA, respectively? Was a quantitative analysis performed? Has the method been validated?
- If there are two spots from GAA, what is the difference? Since they come from the same compound, they should be at the same height in the chromatogram.
- In the conclusions, the authors state that the method can be used for quantitative determinations. So, I will repeat the question: has the method been validated? Is there any previous work that deals with this? It should be quoted, then.
After minor additions and explanations, I recommend the work for printing.
Author Response
Dear Dr
Happy day
First of all we would like to thank you all about your efforts, time and valuable advices which really improved this paper.
It is a pleasure for us to response to your valuables’ comments.
All comments are considered.
Kindly blew you can find our response for each of them.
The paper has been revised and any changes was highlighted by green color.
Moved parts (based on the reviewer request) were highlighted by using underline option.
General modifications are also included:
- Adding new 5 references
- Updating all the references
- Adding “a” and “b” to Figure number 1
- Adding additional parts to the introduction and the conclusion (based on the reviewer request).
- Adding “using standard criteria” in different part of text to attract the attention of the reader to use additional details based on the recommended standard criteria for each part.
Reviewer 1
Comments and Suggestions for Authors
The authors attempted to identify, isolate and purify one of the natural toxic compounds, i.e. gossypol, present in cottonseed. They used some chromatographic methods and also UV/IR analysis. Additionally, they studied the similarity between Egyptian and Chinese varieties of cottonseed. The derivative, i.e. gossypol acetic acid, has many pharmacological applications, so it is significant in pharmacy. It also comes from cottonseed, which the authors write about extensively in the introduction to the publication.
The methodology thoroughly describes sample preparation and each point of analysis in detail.
However, I have some comments:
- on what basis did the authors determine that the spots (Fig. 1) contain 17.5% and 1.02% of GAA, respectively? Was a quantitative analysis performed? Has the method been validated?
Many thanks, The isolated crystals from the bands a and b were isolated from the surface of the TLC, purified and determined by the HPLC as described in details in the Material and methods. The HPLC use the standard sample (GAA) to determine the amount in the injected amount of the sample.
- If there are two spots from GAA, what is the difference? Since they come from the same compound, they should be at the same height in the chromatogram.
Many thanks, They are either racemic gossypol acetic acid, (+)-gossypol triacetic acid or (-)-gossypol triacetic acid. As well as gossypol has more isomers and enantiomers.
For more details kindly refer to Dowd et al (1999), Dowd et al., 2003; Yildirim-Aksoy et al., 2004. .
Dowd, Michael K., Leonard M. Thomas, and Millard C. Calhoun. "Crystal and Molecular Structure of an Enantiomeric Gossypol‐Acetic Acid Clathrate." Journal of the American Oil Chemists' Society 76, no. 11 (1999/11 1999): 1343-50. https://doi.org/10.1007/s11746-999-0148-6.
Dowd, Michael K. "Preparation of Enantiomeric Gossypol by Crystallization." Chirality 15, no. 6 (2003/01 2003): 486-93. https://doi.org/10.1002/chir.10237.
Yildirim-Aksoy, M., C. Lim, M. K. Dowd, P. J. Wan, P. H. Klesius, and C. Shoemaker. "In Vitro Inhibitory Effect of Gossypol from Gossypol-Acetic Acid, and (+)- and (-)-Isomers of Gossypol on the Growth of Edwardsiella Ictaluri." Journal of Applied Microbiology 97, no. 1 (2004/07 2004): 87-92. https://doi.org/10.1111/j.1365-2672.2004.02273.x.
In the result section this has been highlighted in the following sentences
[“Apparently, that could be explained because of the presence of racemic gossypol acetic acid, (+)-gossypol triacetic acid or (-)-gossypol triacetic acid. As well as gossypol has more isomers and enantiomers. For more details refer to Dowd et al., (1999), Dowd (2003) and Yildirim-Aksoy (2004).”]
[“Meanwhile, the GAA in spot a and b did not have identical configurations as explained by Dowd et al., (1999), Dowd (2003) and Yildirim-Aksoy (2004). That could explain its presence in different spots because of different mobility on the TLC.”]
- In the conclusions, the authors state that the method can be used for quantitative determinations. So, I will repeat the question: has the method been validated? Is there any previous work that deals with this? It should be quoted, then.
Many thanks, The paper includes 3 quantitative methods
- Using the direct weighting for measuring any of the prepared gossypol acetic acid crystals. (Has been added to the text) “and then weighted” line. Many thanks for this point
- HPLC analysis
- Determine the amount of the gossypol acetic acid spectrophotometrically and drive its amount from the standard curve [data not shown]
We recommended the HPLC because it give more accurate result. Using the standard curve is not the aim of this study but we demonstrate it to be a handy tool for any one could not access to a modern HPLC.

Reviewer 2 Report
Comments and Suggestions for Authors
The authors described protocol about gossypol acetic acid purification. The paper is quite descriptive and I cannot find its key merit or finding. The described procedure is commonly used.
1, in introduction, quite words obout drug effect, little about technique improvment about chromatographic background.
2, in extraction and purification, no parameters optimization.
3, for figures, add one to illustrate the whole process.
4, in conclusion, what is the specific finding of this work?
Author Response
Dear Dr
Happy day
First of all we would like to thank you all about your efforts, time and valuable advices which really improved this paper.
It is a pleasure for us to response to your valuables’ comments.
All comments are considered.
Kindly blew you can find our response for each of them.
The paper has been revised and any changes was highlighted by green color.
Moved parts (based on the reviewer request) were highlighted by using underline option.
General modifications are also included:
- Adding new 5 references
- Updating all the references
- Adding “a” and “b” to Figure number 1
- Adding additional parts to the introduction and the conclusion (based on the reviewer request).
- Adding “using standard criteria” in different part of text to attract the attention of the reader to use additional details based on the recommended standard criteria for each part.
-
1, in introduction, quite words about drug effect, little about technique improvement about chromatographic background.
Few sentences were added to highlight the importance of the silica gel based method such as the TLC.
“One interesting properties of the GAA its different chemical isomers and enantiomers like (+)-gossypol triacetic acid or (-)-gossypol triacetic acid [44-46].”
“Some compounds, reported by different research groups or from different plants obviously represent the same structures. Meanwhile they showed different Rr values on thin layer chromatography (TLC) [47,48].”
2, in extraction and purification, no parameters optimization.
Many thanks. This work was conducted in a phytochemistry lab. And all the methods are optimized over years. Meanwhile we added this part of phrase in different part of the text “using standard criteria”. We believe that the reader will consult textbooks for any details. And that the paper will be better if such details are not included.
3, for figures, add one to illustrate the whole process.
Many thanks, A and b letters are added to the Figures
4, in conclusion, what is the specific finding of this work?
Many thanks, few sentences are added to the conclusion to cover this point
Many thanks

Reviewer 3 Report
Comments and Suggestions for Authors
After analyzing the document titled “Silica Gel chromatographic methods for identification, isolation and purification of Gossypol acetic acid prepared in lab" for possible publication in the Gels Journal, my observations are the following:
*Title:
- Delete prepared in lab, I consider it not necessary.
*Introduction.
-Define Clearly T/C. line 50
-Review the numbering of bibliographic citations within the text. incorrect numbering, for example, lines 56-58.
*Results and discussion
-Delete lines 205 to 233. This entire part is not results, it is information that can be included in the introduction but is not results.
-Eliminate . on line 239, also: on line 295.
-There is an incomplete bibliography, for example, line 361. I recommend reviewing the magazine's instructions on writing the bibliography.
In general, the document deficiencies further discussion of the results obtained, it is necessary to compare the results with those obtained by other authors, including with the results obtained by other analytical methods.
Considering these weaknesses, I have to recommend a major correction of the manuscript.
Author Response
Dear Dr
Happy day
First of all we would like to thank you all about your efforts, time and valuable advices which really improved this paper.
It is a pleasure for us to response to your valuables’ comments.
All comments are considered.
Kindly blew you can find our response for each of them.
The paper has been revised and any changes was highlighted by green color.
Moved parts (based on the reviewer request) were highlighted by using underline option.
General modifications are also included:
- Adding new 5 references
- Updating all the references
- Adding “a” and “b” to Figure number 1
- Adding additional parts to the introduction and the conclusion (based on the reviewer request).
- Adding “using standard criteria” in different part of text to attract the attention of the reader to use additional details based on the recommended standard criteria for each part
After analyzing the document titled “Silica Gel chromatographic methods for identification, isolation and purification of Gossypol acetic acid prepared in lab" for possible publication in the Gels Journal, my observations are the following:
*Title:
- Delete prepared in lab, I consider it not necessary.
Many thanks, It was deleted
*Introduction.
-Define Clearly T/C. line 50
Many thanks, T/C Mean treated over control tumor volume ratio (T/C%). Kindly referee to https://aacrjournals.org/clincancerres/article/9/11/4227/202621/Clinical-Predictive-Value-of-the-in-Vitro-Cell
We have also added the mean of T/C and clarified it in the text
-Review the numbering of bibliographic citations within the text. incorrect numbering, for example, lines 56-58.
Many thanks, All references are updated and revised. Five additional references were added.
Dowd, Michael K., Leonard M. Thomas, and Millard C. Calhoun. "Crystal and Molecular Structure of an Enantiomeric Gossypol‐Acetic Acid Clathrate." Journal of the American Oil Chemists' Society 76, no. 11 (1999/11 1999): 1343-50. https://doi.org/10.1007/s11746-999-0148-6.
Dowd, Michael K. "Preparation of Enantiomeric Gossypol by Crystallization." Chirality 15, no. 6 (2003/01 2003): 486-93. https://doi.org/10.1002/chir.10237.
Yildirim-Aksoy, M., C. Lim, M. K. Dowd, P. J. Wan, P. H. Klesius, and C. Shoemaker. "In Vitro Inhibitory Effect of Gossypol from Gossypol-Acetic Acid, and (+)- and (-)-Isomers of Gossypol on the Growth of Edwardsiella Ictaluri." Journal of Applied Microbiology 97, no. 1 (2004/07 2004): 87-92. https://doi.org/10.1111/j.1365-2672.2004.02273.x.
Abreo, M. J., and A. T. Sneden. "4-Hydroxy-25-Desoxyneorollinicin, a New Bistetrahydrofuranoid Acetogenin from Rollinia Papilionella.". J. Nat. Prod. 52 (1989): 822-28.
Gu, Z-M. , G-X. Zhao, N. H. Oberiies, L. Zeng, and J. L. McLaughlin. "Chapter 11: Recent Advances in Phytochemistry- Phytochemistry of Medicinal Plants-Volume 29." edited by J. T. Arnason, Mata, R., Romeo, J. T. , 249-310: SPRINGER SCIENCE+BUSINESS MEDIA, LLC, 1995.
*Results and discussion
-Delete lines 205 to 233. This entire part is not results; it is information that can be included in the introduction but is not results.
Many thanks, part has been transferred to the introduction part (lines might not equal in this version).
However, some sentences are left to prepare the reader to get the mean of the paper during reading the “result and discussion parts”
The moved part to the introduction are highlight by use the underline option. Again repeated sentences or meaning were removed
Any other style or grammar correction is highlighted by green color
-Eliminate . on line 239, also: on line 295.
Many thanks, “Eliminate” as word has been changed to “remove”
-There is an incomplete bibliography, for example, line 361. I recommend reviewing the magazine's instructions on writing the bibliography.
Many thanks, All references were updated. So sorry. In many cases it was the Endnote who did not show all the loaded data like the book’s editors.
In general, the document deficiencies further discussion of the results obtained, it is necessary to compare the results with those obtained by other authors, including with the results obtained by other analytical methods.
Five references are added to show the concept behind using the TLC in the analysis. Sentences are highlighted when to use the TLC and when to use the HPLC.
Many thanks
With my pleasure

Round 2
Reviewer 2 Report
Comments and Suggestions for Authors
This is not much improved. Therefore, I insist my former decision.
Author Response
Dear Prof. Dr.
Happy day.
Many thanks that you put a load on me to improve this paper. I am thankful for that.
Kindly concern that
Text highlighted with green color was changed in the last revision
Text highlighted with yellow color is changed in this revision.
Text underline is moved from place to place without modifications.
Concerning your suggestions
1, in introduction, quite words about drug effect, little about technique improvement about chromatographic background.
Many thanks for your kind suggestion
We have added a hint about the purpose of inventing the open column system “thin layer chromatography” which invented by Izmailov and Shraiber at 1938. The main problem which forces scientists to search for an alternative to the column chromatography is the mobility of the solvent. Izmailov and Shraiber perfectly open the get using to the thin layer chromatography. Stahl the man who optimize the thin layer chromatography has been described as well. In the past version we highlighted that TLC can help in separating enantiomers, Additional references are added.
2, in extraction and purification, no parameters optimization.
Many thanks for your kind suggestion
We have adding the Rf as a parameter for calculating the separation process. Additionally, we have highlighted that optimizing the chamber saturation give linear results where all spots are located in the same line. Many additional details have been considered. We found that it will give more information to the reader. For example, all extraction and experimental conditions done in temperature less or equal to 40oC to avoid the degradation of the samples. The crystallization steps done on darkness. By revising many papers, we found that the amount of total gossypol in the cottonseed’s variates are nearly equal to our finding and reference was add to support that.
3, for figures, add one to illustrate the whole process.
Many thanks for your kind suggestion
One diagram was added to summarize the process. Where the silica gel-based methods are highlighted to show their importance in identifying gossypol acetic acids and its expected enantiomers.
4, in conclusion, what is the specific finding of this work?
Many thanks for your kind suggestion
The conclusion part is completely changed to highlight the specific finding which is that the silica gel-based method like the TLC could separate the similar structure (e.g., enantiomers or non-complete structures). Even the main aim of this study is to help the scientist interesting to isolate the gossypol as gossypol acetic acid in the lab to conduct their own experiment, but after following your kind suggestion points as well the other reviewer point, it become a complete study combine between the phytochemistry and the chromatography.
Finally, I would like to thank you a lot for your kind support suggestion and interest in this paper

Reviewer 3 Report
Comments and Suggestions for Authors
Dear Editor,
In the present form of the document, and after the corrections made, I consider that the document meets the requirements to be accepted for publication in the journal.
Author Response
Dear Dr
Happy day
Thank you very much for your kind revising for the paper and for the important points you have kindly suggested to improve it. This has made us feel deeply grateful for your help.
With our pleasure